# Risk Factors for Poor Sleep Quality and Subjective Cognitive Decline in Older Adults Living in the United States During the COVID-19 Pandemic

**DOI:** 10.3390/healthcare13060675

**Published:** 2025-03-20

**Authors:** Celina Pluim McDowell, Jairo E. Martinez, Averi Giudicessi, Diana Munera, Clara Vila-Castelar, Edmarie Guzmán-Vélez, Liliana Ramirez-Gomez, Jeanne F. Duffy, Alice Cronin-Golomb, Yakeel T. Quiroz

**Affiliations:** 1Department of Psychological and Brain Sciences, Boston University, Boston, MA 02215, USA; 2Department of Psychiatry, Massachusetts General Hospital, Harvard Medical School, Boston, MA 02114, USA; 3Division of Sleep and Circadian Disorders, Department of Medicine, Brigham and Women’s Hospital, Boston, MA 02115, USA; 4Division of Sleep Medicine, Harvard Medical School, Boston, MA 02115, USA; 5Department of Neurology, Massachusetts General Hospital, Harvard Medical School, Boston, MA 02114, USA

**Keywords:** sleep quality, subjective cognitive decline, mild cognitive impairment, depression, anxiety, COVID-19

## Abstract

Background/Objectives: Poor sleep quality, depression, and anxiety are associated with subjective cognitive decline (SCD) and greater risk for developing cognitive impairment and Alzheimer’s disease. The COVID-19 pandemic negatively impacted sleep habits and psychological well-being in many individuals, yet their relationship to SCD in this context remains understudied. We examined sociodemographic characteristics, depression, anxiety, and sleep changes during the pandemic (i.e., increased problems/poorer quality) and SCD in older individuals living in the US. Methods: In total, 288 older adults (M_age_ = 67.4 [7.4]) completed an online survey including a sociodemographic questionnaire, the Center for Epidemiologic Studies Depression Scale (Revised) (CES-D-10; depression), General Anxiety Disorder-7 (GAD-7; anxiety), the Everyday Cognition scale (ECog; SCD), and a question about increased sleep problems/worsened sleep quality during the pandemic. Hierarchical and logistic regression analyses were used to assess relations among sociodemographic factors, depression, anxiety, changes in sleep quality, and SCD. Results: Self-reported pandemic-related impairments in sleep were associated with greater SCD (β = 0.16, *p* = 0.01). Depression (β = 0.46, *p* < 0.001) and anxiety (β = 0.29, *p* < 0.001) were also associated with greater SCD. Depression (OR = 1.17, *p* < 0.001) and anxiety (OR = 1.15, *p* = 0.017) predicted reported poorer sleep during the pandemic. Conclusions: Depression, anxiety, and poorer sleep quality during the COVID-19 pandemic were associated with greater SCD concerns. Greater depression and anxiety were also associated with the reported sleep problems/worsened sleep quality. Prevention and management of anxiety and depressive symptoms may help maintain sleep quality and reduce risk of cognitive decline.

## 1. Introduction

In 2020, the outbreak of the Coronavirus disease 2019 (COVID-19) prompted local and federal stay-at-home and social distancing orders to minimize person-to-person contact and reduce the spread of the virus. These safety measures disrupted regular work and life routines for many people during the global pandemic, negatively affecting psychological well-being, including increased feelings of depression and anxiety, and reductions in sleep quality [1,2,3]. Older adults, already at a higher risk than younger adults of experiencing worsened sleep quality [4], were especially impacted by the pandemic in these domains [5].

Poor sleep quality increases the risk of objective cognitive decline in older adults [6], characterized by objective impairment on standardized cognitive tests. Poor sleep quality is also associated with subjective cognitive decline (SCD) [7,8], the self-reported experience of worsening cognitive abilities [9], itself a known risk factor for objective cognitive decline and developing mild cognitive impairment (MCI) and Alzheimer’s disease (AD). Given that SCD is often endorsed before the onset of objective cognitive decline [9], understanding the factors associated with SCD is pertinent to reduce overall cognitive decline risk for older adults. Few studies have examined associations between sleep quality and subjective cognitive functioning during the pandemic in older adults, yielding mixed results. One study of older adults in the United Kingdom found that poorer sleep quality was associated with greater concerns about cognitive functioning [10], while a study in Belgium found no associations between subjective cognition and sleep disturbances [11]. Overall, there is a paucity of research examining sleep quality and SCD during the pandemic, particularly for older adults living in the United States (US), and there is a need to better understand the role of psychological well-being in these relations.

Psychological well-being is relevant to cognitive and sleep health. Symptoms of depression and anxiety, both independent risk factors for cognitive decline [12,13], were associated with greater SCD during the pandemic [11,14]. Depression and anxiety are also known to worsen sleep quality in older adults [5,11,15]. In addition to psychological factors, socioeconomic factors such as lower income, financial hardship, fewer years of education, and unemployment have been associated with worse sleep quality in older adults prior to [16,17] and during the pandemic [18]. These associations were proposed to result from higher stress, economic burden, and depression [16,18]. More research is needed to better understand the relations among sociodemographic factors, sleep quality, psychological well-being, and SCD in older adults. Such research may direct efforts toward tailored treatment or preventative approaches for those at the greatest risk of poor sleep outcomes and at the greatest risk of cognitive decline and AD. Examining these associations in the context of COVID-19 is of particular importance, because of the pandemic’s negative impact on economic security, sleep quality, and psychological well-being for older adults [2,3,19].

This study sought to examine associations among sociodemographic characteristics, depression, anxiety, SCD, and sleep changes during the COVID-19 pandemic in older individuals in the US. We hypothesized that higher SCD ratings would be associated with negative sleep changes (increased problems/poorer quality), and greater depression and anxiety during the pandemic would strengthen this association in those experiencing sleep changes. In addition, we expected that lower education levels, lower self-reported income, greater depressive symptoms, and greater anxiety ratings would be associated with negative changes in sleep quality.

## 2. Materials and Methods

### 2.1. Participants

The initial sample in this study included 501 survey respondents living in the US. Due to low participation among Native American (*n* = 5) and Native Hawaiian/Pacific Islander (*n* = 2) participants, these individuals were not included in the analyses. Moreover, 14 participants missing ethnoracial identity, 23 participants who reported “other”, and 2 participants who reported “prefer not to answer” for ethnoracial identity were also excluded from the analyses, as ethnoracial identity was required for between-group comparisons and was planned to be included in analyses examining the associations of sociodemographic factors and negative changes in sleep. For other participants missing data pertinent to the analyses (i.e., missing items from depression, anxiety, or SCD measures), listwise deletion was implemented (Figure 1). The final sample of participants included 288 individuals (average age = 67.4; SD = 7.4; range = 55–90; 75% female) living in the US, who self-identified as Caucasian/White (*n* = 227), Hispanic/Latino (*n* = 23), African American/Black (*n* = 22), or Asian (*n* = 16). The participants, on average, had a college-level education (17.7 years; SD = 2.3; range = 5–20). Participants in this ancillary study were part of an international study investigating well-being and cognition in older adults during the COVID-19 pandemic. Participants were recruited through social media, virtual meetings, and investigator contacts, comprising a non-probabilistic, convenience sample [20]. They completed a one-hour survey online including questionnaires on demographics, psychological well-being (i.e., depression, anxiety), sleep changes, and SCD. Participants provided informed consent online, and data were collected and stored using Research Electronic Data Capture (REDCap). All study procedures were approved by the Mass General Brigham Institutional Review Board.

### 2.2. Measures

Sociodemographic factors. Self-reported income options were “low income”, “middle income”, and “high income”. Marital status was dichotomized to indicate married or unmarried. Occupational status was dichotomized to indicate employed or not employed.

Sleep changes. The online survey included the Epidemic Pandemic Impacts Inventory (EPII) [21], a questionnaire assessing the impact of the COVID-19 pandemic on several domains of personal, family, and professional life. One item from the emotional health and well-being domain assessed changes in sleep quality (“Since the coronavirus disease pandemic began, what has changed for you or your family? … Increase in sleep problems or poor sleep quality”.) and participants responded with “yes (me)”, “yes (person at home)”, “no”, or “not applicable”. This item was dichotomized to reflect changes in the individual’s sleep quality (yes (me) or no) for the analyses.

Depression and Anxiety. The Center for Epidemiologic Studies Depression Scale (Revised) (CES-D-10) [22] was used to assess symptoms of depression. Respondents answer 10 items about how often they experienced symptoms over the past week (e.g., “I felt depressed”), with answers ranging from 0 (“rarely or none of the time”) to 3 (“all of the time”). Positively worded items are reverse-scored. Higher scores indicate greater depression. Total scores ≥ 10 indicate clinically significant depression (possible range: 0–30). The General Anxiety Disorder-7 (GAD-7) [23] was used to measure symptoms of anxiety. Respondents answer 7 items assessing how often they were bothered by anxiety symptoms over the past two weeks (e.g., “Feeling nervous, anxious, or on edge”), with responses ranging from 0 (“not at all”) to 3 (“nearly every day”). Responses were summed for a total score, and higher scores indicate greater anxiety. Scores of 5, 10, and 15 indicate mild, moderate, and severe anxiety, respectively (possible range: 0–21). Both the CES-D-10 and the GAD-7 have demonstrated good reliability and validity, including validity for detecting depression and anxiety, respectively, in older adults [22,23,24].

**Figure 1 healthcare-13-00675-f001:**
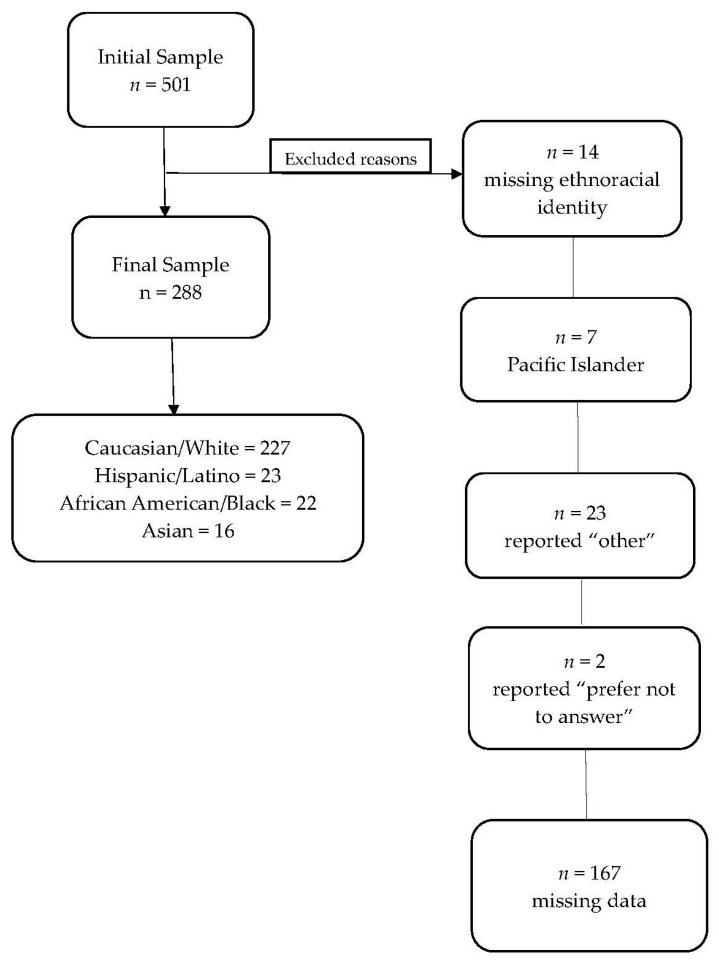
Study flowchart of participants.

Subjective Cognitive Decline. The Measurement of Everyday Cognition (ECog) [25] is a 39-item questionnaire measuring SCD. Items map onto cognitive domains of memory, language, visuospatial abilities, and executive function. Participants compare current abilities with 10 years ago (e.g., “remembering where I have placed objects”) using a scale from 1 (“better or no change”) to 4 (“consistently much worse”). The total score is calculated by taking the average score of all items answered (possible range: 1–4), with higher total scores indicating greater SCD.

### 2.3. Analyses

One-way ANOVA and chi-squared tests were first conducted to examine demographic differences among ethnoracial groups. A multiple regression analysis was used to assess the relation between changes in sleep and SCD, including change in sleep quality (coded as no increase in sleep problems or poor sleep quality = 0, and yes for an increase in sleep problems or poor sleep quality = 1) as a predictor and ECog total score as the criterion. Pearson’s correlation analyses were conducted to determine whether sociodemographic covariates (age, education) should be included as predictors in this regression analysis. Two separate hierarchical regressions were also conducted to examine whether changes in sleep moderated the respective associations of depression with SCD and anxiety with SCD. In each regression, covariates (age, education) were entered in Step 1, change in sleep quality and CES-D-10 score (or GAD-7 score) were entered in Step 2, the interaction term of change in sleep quality*CES-D-10 (or change in sleep quality*GAD-7) was entered in Step 3, and ECog total score was the criterion. Logistic regression was used to assess predictors of changes in sleep quality, including depression (CES-D-10), anxiety (GAD-7), and sociodemographic factors (i.e., age, sex, education, ethnoracial group, income, marital status, occupational status); Pearson’s chi-squared tests were conducted to determine which categorical sociodemographic factors to include as predictors in this regression analysis.

## 3. Results

Demographic variables, SCD, depression, and anxiety scores were within acceptable limits of normality [26]; skewness was <1.7 and kurtosis was ≤3 for all variables. On average, White participants were older than Latino participants (F(3, 284) = 3.41, *p* = 0.02, eta^2^ = 0.04). There were more unemployed participants in the White group (*X*^2^(3, N = 288) = 12.9, *p* = 0.005, *V* = 0.21). There were no group differences in years of education, sex distribution, marital status distribution, income distribution, distribution of participants reporting sleep changes, depression scores, anxiety scores, or ECog scores (all *p*-values > 0.10). Average depression scores were below the cutoff measure for clinically significant depression (average = 7.94, SD = 6.09, range = 0–27), average anxiety scores were below the cutoff measure for mild anxiety (average = 3.61, SD = 4.42, range = 0–21), and average ECog total scores were below the suggested cutoff for MCI [27] (average = 1.38; SD = 0.42; range = 1.00–3.01). In total, 120 participants reported an increase in sleep problems or poor sleep quality (Table 1).

Sleep and SCD. Correlation analyses revealed a significant association between changes in sleep problems or quality and SCD (*r* = 0.14, *p* = 0.02). Correlations of age (*r* = 0.07, *p* = 0.24) and education (*r* = −0.04, *p* = 0.48) with SCD were not significant; therefore, these factors were not included as covariates in the regression. The regression examining the association between changes in sleep problems or sleep quality and SCD indicated that those who endorsed an increase in sleep problems or poor sleep quality during the pandemic had higher ECog scores (F(1, 286) = 5.92, *p* = 0.02, *R*^2^ = 0.02; β = 0.14, *p* = 0.02). Model 2 of the hierarchical regression examining changes in sleep and CES-D-10 scores as predictors of SCD was significant, indicating that higher depression scores predicted greater SCD (F(4, 283) = 16.70, *p* < 0.001, *R*^2^ = 0.19; β = 0.46, *p* < 0.001). The overall final model of this regression examining the interaction between changes in sleep and CES-D-10 scores was significant, although the interaction term itself was not a significant predictor of SCD (F(5, 282) = 14.10, *p* < 0.001, *R*^2^ = 0.20; β = −0.15, *p* = 0.08). Model 2 of the hierarchical regression examining changes in sleep and GAD-7 scores as predictors of SCD was significant, indicating that higher anxiety scores predicted greater SCD (F(4, 283) = 7.35, *p* < 0.001, *R*^2^ = 0.09; β = 0.29, *p* < 0.001). The final model of this regression including the interaction between changes in sleep and GAD-7 scores was significant, indicating that the association between anxiety and SCD was stronger in those with no changes in sleep problems or quality (F(5, 282) = 7.60, *p* < 0.001, *R*^2^ = 0.12; β = −0.31 *p* = 0.005; Figure 2).

Predictors of change in sleep quality. A higher proportion of women reported changes in sleep quality compared with men (*X*^2^(1, N = 288) = 6.70, *p* = 0.01, *V* = 0.15). There were no differences in reported income level (*X*^2^(2, N = 288) = 0.79, *p* = 0.67, *V* = 0.05), marital status (*X*^2^(1, N = 288) = 0.02, *p* = 0.88, *V* = 0.01), or employment status (*X*^2^(1, N = 288) = 0.04, *p* = 0.85, *V* = 0.01) in those who reported sleep changes compared with those who reported no sleep changes. Therefore, income, marital status, and employment status were not included as predictors in the logistic regression. The overall model of the logistic regression assessing depression, anxiety, and demographic predictors (age, sex, education) of changes in sleep problems or sleep quality was significant (*X*^2^(5) = 85.43, *p* < 0.001, Nagelkerke *R*^2^ = 0.35). Greater depression (OR = 1.15, *p* < 0.001, 95% CI [1.07, 1.24]) and greater anxiety (OR = 1.15, *p* = 0.009, 95% CI [1.04, 1.29]) were associated with a greater likelihood of reporting increased sleep problems or poor sleep quality during the pandemic. No other sociodemographic predictors (age, sex, education) were associated with increased sleep problems/poor sleep quality (Table 2).

## 4. Discussion

This study examined negative changes in sleep quality (i.e., increased sleep problems or poor sleep quality) during the COVID-19 pandemic, and their association with sociodemographic factors, depression, anxiety, and SCD in older adults via an online survey. In total, 41% of participants (*n* = 120) reported worsened sleep quality since the beginning of the pandemic. Self-reported increased sleep problems or poor sleep quality during the pandemic and greater depressive symptoms were associated with higher SCD. Greater anxiety ratings were associated with higher SCD ratings, particularly for those who did not report changes in sleep problems or poor sleep quality. We also found that greater depression and greater anxiety ratings were predictors of reported changes in sleep.

Our findings are in line with those of prior studies demonstrating that poor sleep quality is associated with SCD in older adults [7,8], which was not surprising, given that poor sleep quality negatively impacts cognitive functioning and is a risk factor for objective cognitive decline, mild cognitive impairment, and Alzheimer’s disease [6]. We did not quantify sleep quality with standard sleep questionnaires or inquire about the presence or absence of specific sleep problems or sleep disorders; we explicitly asked about increased sleep problems or poor sleep quality during the pandemic. Such reported changes were associated with higher SCD ratings, suggesting that negative changes in sleep quality may be a risk factor for greater SCD and, consequently, cognitive decline. It is generally worth assessing changes in sleep quality or sleep problems in older adults (beyond simply using a cutoff score to determine if one has good or poor sleep quality), as assessing such changes may help identify those experiencing cognitive concerns by prompting further SCD inquiries. For example, older adults raising concerns about changes to sleep quality to primary care providers or mental health care providers could prompt these providers to also query about subjective cognitive changes and/or refer them for further assessment to intervene early and prevent objective cognitive decline. Likewise, in older adults with SCD concerns, identifying potential negative changes in sleep quality could highlight modifiable targets associated with SCD and future cognitive decline risk. For example, individuals who present for a neuropsychological evaluation due to SCD concerns are typically asked about their sleep generally and may be administered a screening measure of sleep quality or specific sleep disorders. However, our findings highlight that it is worth asking these individuals specifically about changes to sleep quality, with an understanding that endorsing negative changes in sleep quality may illuminate potential targets (e.g., implementing sleep hygiene or relaxation strategies) to improve sleep quality, improve SCD, and prevent further cognitive decline, even if reported sleep issues do not rise to the levels of a sleep disorder according to the screening measure’s cutoffs. Assessing sleep quality or sleep problems over time may be especially useful to identify such individuals at risk for future cognitive decline. More longitudinal research is needed to better characterize these associations.

Although average depression and anxiety ratings were subclinical, greater symptoms of depression and anxiety were both associated with greater SCD, in line with prior work demonstrating these associations [12,13]. When examined together with sleep changes, only depression and anxiety ratings—not sleep changes—predicted SCD. There was no interaction effect of depression and sleep changes on SCD ratings. Surprisingly, we found an interaction of anxiety ratings and sleep changes on SCD, such that the association between anxiety ratings and SCD was stronger in those who reported no changes in sleep quality. These findings suggest that depression and anxiety are distinct predictors of SCD, independent of reductions in sleep quality or increases in sleep problems. Past work has demonstrated that while both physical health factors (including sleep disturbances) and psychological factors correlate with subjective cognitive concerns, associations with psychological factors, particularly depression and anxiety, were strongest, indicating that psychological well-being is especially pertinent to subjective cognitive functioning [28]. Additional studies have found that even subclinical symptoms of depression and anxiety may have a greater influence on SCD ratings in older adults compared with physical health conditions and self-reported sleep quality [29]. The stronger association of anxiety ratings with SCD in those without sleep changes was unexpected, and this result may have been influenced by the context of the pandemic. For example, research has demonstrated fluctuations in psychological well-being and subjective cognition throughout the course of the pandemic due to changes in governmental regulations and social distancing policies, in line with surges or declines in infection rates [30]. In our cross-sectional study, we were not able to determine how these well-being ratings may have changed throughout the pandemic, though it is possible that fluctuations over time influenced these associations. Taken together, these findings suggest that anxiety and depression, even at subclinical severity ratings, may be more important risk factors of SCD than changes in sleep quality or sleep problems. Studies examining these factors concurrently with SCD are limited, particularly within the context of the COVID-19 pandemic, and more longitudinal work is needed to better understand the directionality of these associations overall.

Only depression and anxiety were significant predictors of an increase in sleep problems or poor sleep quality. Interestingly, no sociodemographic factors were associated with changes in sleep, contrary to past work examining these associations [16,17,18]. This discrepancy could be due to the nature of our sample, who were, on average, highly educated and mostly self-reported middle income, and had less variability in education and income levels than the samples included in other studies. More research is needed with samples that are more heterogeneous in education and income levels. By contrast, our findings are consistent with prior studies demonstrating associations between psychological well-being and sleep quality in older adults during the pandemic [5,11,15], suggesting that managing depression and anxiety is important for maintaining sleep health as well as cognitive health. Targeting symptoms of depression and anxiety through psychotherapy and/or medication management may concurrently improve psychiatric symptoms, lead to improvements in sleep quality, and reduce the cognitive decline risk in older adults. Those endorsing symptoms of depression and/or anxiety should also be screened for changes to sleep quality. Doing so may allow providers to implement a multi-pronged approach to reduce the overall risk of cognitive decline by concurrently targeting both sleep quality and psychiatric symptoms. There is evidence that targeting sleep quality as part of cognitive behavioral therapy (CBT) for depression and anxiety improves both sleep and psychiatric outcomes [31]; however, more research is needed to examine the cognitive outcomes of such treatments.

Limitations. This study has limitations. The cross-sectional design impacts interpretation of the associations among sociodemographic factors, psychological well-being, sleep quality, and SCD, in that we are not able to infer causality. More longitudinal research in this area is needed to examine these associations and their directionality. Because the present study was conducted online, there may be enrollment bias for participants who had access to the internet and a device to complete online questionnaires, which could have impacted the ethnoracial and socioeconomic diversity of our sample. Our sample had subclinical ratings of depression and anxiety on average, which may limit the ability to generalize to older adults with more severe anxiety or depressive symptoms. Research on larger samples is needed to examine sociodemographic factors, psychological well-being, and their relation to sleep quality in more representative, ethnoracially diverse groups. Future work should also consider including objective measures of cognition and a more comprehensive assessment of sleep quality to assess the impact of sleep quality on cognitive decline and the roles of depression, anxiety, and other aspects of psychological well-being in these associations.

## 5. Conclusions

Overall, our findings suggest that changes in sleep quality (i.e., worsened quality, more sleep problems) associated with the COVID-19 pandemic may constitute a risk factor for SCD or future objective cognitive decline. Further, we found that symptoms of depression and anxiety may be risk factors both for SCD and negative changes in sleep quality or more sleep problems during the pandemic. Inquiring about changes in sleep quality and experiences of depression or anxiety may help providers identify older adults who are also experiencing SCD or who are at risk of future cognitive decline. Likewise, for older adults endorsing SCD, assessing for recent changes to sleep quality and symptoms of depression or anxiety may help identify potential treatment targets to reduce SCD and cognitive decline risk overall. Strategies to treat or combat anxiety and depression in older adults such as psychotherapy, behavioral interventions, and/or medication management may help maintain better sleep quality and reduce sleep problems in older adults and may help reduce the risk of SCD and cognitive decline.

## Figures and Tables

**Figure 2 healthcare-13-00675-f002:**
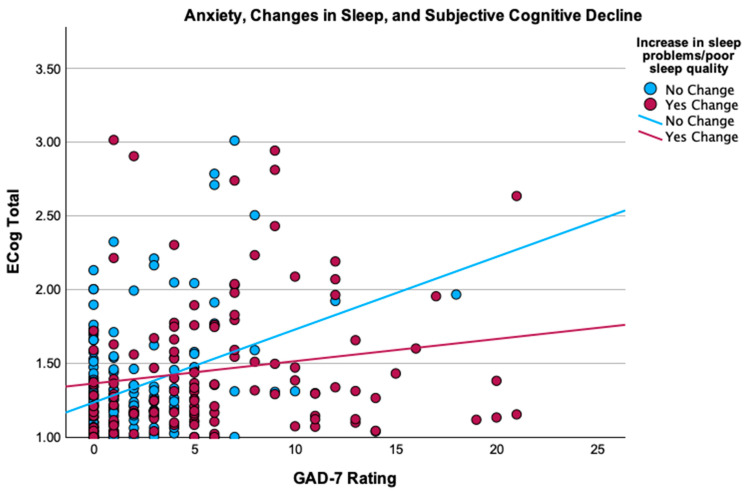
Anxiety ratings, changes in sleep (increased problems/poor sleep quality), and subjective cognitive decline. Higher anxiety ratings (GAD-7) were associated with higher SCD ratings (ECog). This association was moderated by a reported increase in sleep problems or poor sleep quality, such that this association was stronger in participants who did not report increases in sleep problems or poor sleep quality during COVID-19.

**Table 1 healthcare-13-00675-t001:** Sample demographic information.

	Total Sample (*n* = 288) Mean (SD)	White (*n* = 227) Mean (SD)	Latino (*n* = 23) Mean (SD)	Black (*n* = 22) Mean (SD)	Asian (*n* = 16) Mean (SD)	*p*
Age (years)	67.41 (7.43)	67.97 (7.19)	63.00 (6.38)	66.23 (7.93)	65.76 (7.67)	0.02
Education (years)	17.68 (2.31)	17.63 (2.36)	17.91 (1.95)	17.14 (2.34)	18.75 (1.77)	0.17
Sex (male:female)	73:215	62:165	3:20	3:19	5:11	0.24
Marital status (married:unmarried)	95:193	79:148	6:17	8:14	2:14	0.26
Income (low/middle/high)	32/200/56	23/157/47	3/16/4	5/15/2	1/12/3	0.57
Occupational status (employed:not employed)	87:201	66:161	14:9	4:18	3:13	0.005
Sleep changes (yes:no)	120:168	86:138	15:8	10:12	6:10	0.11
CES-D-10	7.94 (6.09)	8.12 (6.13)	7.39 (5.19)	7.45 (5.83)	6.88 (7.23)	0.80
GAD-7	3.61 (4.42)	3.74 (4.50)	3.57 (3.59)	2.77 (3.99)	3.00 (5.13)	0.73
ECog	1.38 (0.42)	1.39 (0.44)	1.35 (0.33)	1.34 (0.28)	1.29 (0.29)	0.87

Note. CES-D-10, Center for Epidemiologic Studies Depression Scale (Revised); GAD-7, General Anxiety Disorder-7; ECog, Everyday Cognition Scale.

**Table 2 healthcare-13-00675-t002:** Depression, anxiety, and sociodemographic factors as predictors of increases in sleep problems or poor sleep quality.

	B	SEB	Wald	*p*-Value	OR	95% CI
Age (years)	−0.001	0.02	0.004	0.95	1.00	0.96–1.04
Sex	0.39	0.35	1.23	0.27	1.47	0.74–2.91
Education (years)	0.08	0.06	1.5	0.22	1.08	0.936–1.22
CES-D-10	0.14	0.04	14.09	<0.001	1.15	1.07–1.24
GAD-7	0.14	0.06	6.74	0.009	1.15	1.04–1.29

Note. OR, odds ratio; 95% CI, 95% confidence interval. Males were the reference group for sex. CES-D-10, Center for Epidemiologic Studies Depression Scale (Revised); GAD-7, General Anxiety Disorder-7.

## Data Availability

The data that support the conclusions of this article are available from the corresponding author upon reasonable request.

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
