# Peer review of "Risk Factors for Poor Sleep Quality and Subjective Cognitive Decline in Older Adults Living in the United States During the COVID-19 Pandemic"

_healthcare, 2025, doi:10.3390/healthcare13060675_

Round 1

Reviewer 1 Report

Comments and Suggestions for Authors

Thank you for the opportunity to review the paper titled " Risk Factors for Poor Sleep Quality and Subjective Cognitive Decline in Older Adults Living in the United States during the COVID-19 Pandemic." I have listed my suggestions below, and I hope they will strengthen the research.

Introduction:

It would be better if you could explain more clearly the concepts of objective cognitive decline and subjective cognitive decline. Are they measured differently? Why did the study choose to focus on subjective cognitive decline?"

Materials & Methods

The sentence is difficult to understand “Participants were part of a parent international study investigating well-being and cognition in older adults during the COVID-19 pandemic”, what is the “a parent international study”?

Methods      

Measure

When introducing the instruments used to measure such as depression and anxiety, please include information about their reliability and validity to ensure the instruments are rigorous.

Missing data handling was not addressed. The manuscript does not specify whether missing data were handled through listwise deletion, multiple imputation, or another method. The absence of this information raises concerns about potential bias, as unaddressed missing data can distort statistical estimates and reduce the reliability of the findings.

The authors did not perform t-tests or ANOVAs to determine whether independent variables were significantly associated with subjective cognitive decline before including them in the regression model. Without this step, it is unclear if all included predictors are relevant, increasing the risk of including non-significant variables that do not contribute to the model.

No Pearson chi-square test was conducted before logistic regression. Categorical predictors were included in the logistic regression model without first assessing their bivariate association with sleep problems. The omission of chi-square tests raises concerns about whether these variables are meaningful contributors to the model.

Discussion

Based on the findings, what specific suggestions and recommendations can be made for practical application? For instance, how can the results be translated into actionable strategies or interventions in real-world settings? Are there particular practices, policies, or guidelines that should be implemented to address the issues identified in the study? Additionally, what steps can stakeholders, practitioners, or policymakers take to ensure the findings are effectively utilized to achieve meaningful outcomes?

Reviewer 2 Report

Comments and Suggestions for Authors

The paper by McDowell et al. explores the topic of association between poor sleep quality and subjective cognitive impairment in older adults from the US during the COVID-19 pandemic. The methodology is well described and the results are clearly presented. The main problem that limits the value of this finding is that causal relationship between subjective cognitive decline and worse sleep quality cannot be proven with this methodology (it may be result of worse anxiety and depression), but by reading this manuscript authors clearly explained this problem and still this work provides some new insight to the topic.

Here are some suggestions to improve this work:

  1. In Table 1, it would be more informative to show the present value next to the number of people (since the visible differences are largely due to the size of the groups).
  2. Where there are differences between races. If such an analysis was done, please discuss briefly and add the p-value as an additional column in Table 1.
  3. Please add units to Table 2.
  4. Please indicate what sampling methods were used in this study.

Author Response

please see attached word document for our responses.

Reviewer 3 Report

Comments and Suggestions for Authors

The research article has discussed the risk factors associated with poor sleep quality and subjective cognitive decline in older adults living in the United States during the COVID-19 pandemic.

Recommendation

  1. The work is well-designed and organized.
  2. Results and discussion are well corroborated.
  3. The concluding remark is well written.

Minor suggestions

  1. I believe adding a PRISMA flowchart to illustrate the inclusion and exclusion criteria would be beneficial for readers.
  2. Lines, ‘There was no interaction effect of depression …………. well-being is especially pertinent to subjective cognitive functioning [27], can be supported by more published articles, if available to make a strong impact and further can be elaborated by the author's perspectives
  3. Discussion could be elaborated in terms of correlation among psychological factors, cognition decline, and sleep quality based on previously published papers.

Round 2

Reviewer 1 Report

Comments and Suggestions for Authors

Thanks for the responses to my questions.